# Microbial–Metabolomic Exploration of Tea Polyphenols in the Regulation of Serum Indicators, Liver Metabolism, Rumen Microorganisms, and Metabolism in Hu Sheep

**DOI:** 10.3390/ani14182661

**Published:** 2024-09-12

**Authors:** Haibo Wang, Jinshun Zhan, Shengguo Zhao, Haoyun Jiang, Haobin Jia, Yue Pan, Xiaojun Zhong, Junhong Huo

**Affiliations:** 1Institute of Animal Husbandry and Veterinary, Jiangxi Academy of Agricultural Science, Nanchang 330200, China; wanghaibo8815@163.com (H.W.); zhanjinshun1985@163.com (J.Z.); jianghaoyun1995@163.com (H.J.); jiahaobin@jxaas.cn (H.J.); py13782525871@163.com (Y.P.); zhongcaoyangchu@163.com (X.Z.); 2Jiangxi Province Key Laboratory of Animal Green and Healthy Breeding, Institute of Animal Husbandry and Veterinary, Jiangxi Academy of Agricultural Science, Nanchang 330200, China; 3College of Animal Science and Technology, Gansu Agricultural University, Lanzhou 730070, China; zhaosg@gsau.edu.cn; 4College of Animal Science and Veterinary Medicine, Tianjin Agricultural University, Tianjin 300384, China

**Keywords:** tea polyphenols, multi−omics, immune, antioxidant, fermentation, sheep

## Abstract

**Simple Summary:**

Plant extracts have emerged as a viable alternative to antibiotics in livestock production, contributing to the sustainable development of modern animal husbandry. Among these extracts, tea polyphenols, derived from tea leaves as polyhydroxyphenolic compounds, exhibit diverse bioactive properties. This study aimed to assess the impact of tea polyphenols on serum parameters, rumen microbiota, rumen metabolism, and liver metabolism in Hu sheep. The results indicated that dietary supplementation with tea polyphenols did not significantly affect the serum physiological indices of the sheep but enhanced serum immunity and antioxidant levels. Moreover, Firmicutes dominated the network map of the top 80 abundant microorganisms at the genus level, identifying 13 biomarkers at the genus level. Furthermore, tea polyphenols have been found to modulate both rumen and liver metabolism, particularly in relation to energy and lipid metabolism. Furthermore, strong correlations were observed between liver and rumen metabolites, particularly between rumen succinic acid and liver alanyl−serine and methylmalonic acid. This establishes a theoretical basis for tea polyphenol applications and suggests directions for optimizing future additive dosages.

**Abstract:**

This study investigated the impact of tea polyphenols on serum indices, rumen microorganisms, rumen metabolism, and liver metabolism in Hu sheep. Sixty healthy lambs, aged three months and with similar average weights, were chosen and randomly assigned to control (CON), TP400, TP800, and TP1200 groups, each consisting of fifteen lambs. The control group received a basal diet, while the experimental groups were provided with basal diet supplemented with 400 mg/kg, 800 mg/kg, and 1200 mg/kg of tea polyphenols, respectively. Compared with the CON group, the addition of tea polyphenols to the diet significantly increased serum IgA, GSH−Px, and TSOD. In addition, tea polyphenols were able to increase rumen pH but had no significant effect on the rumen NH_3_−N, VFA molar content, and the microbial top 10 phylum and genus levels. Moreover, Firmicutes predominated in the network map of the top 80 abundant microorganisms at the genus level, identifying 13 biomarkers at the genus level. In addition, strong correlations were observed between liver and rumen metabolites, particularly between rumen succinic acid and liver alanyl−serine and methylmalonic acid. Furthermore, tea polyphenol additions changed the enrichment of liver and rumen metabolites in the top five KEGG metabolic pathways, but 400−1200 mg/kg additions had no negative impact on the liver and rumen. In summary, TP significantly influences rumen and liver metabolites in Hu sheep, enhancing lamb immunity and antioxidant capacity, with 400 mg/kg being the most effective dosage.

## 1. Introduction

The maintenance of animal health is a crucial aspect of livestock production. In recent years, researchers have explored the utilization of plants as fundamental resources, employing physical or chemical techniques to extract individual or combined bioactive compounds from plants (e.g., flavonoids, polyphenols, polysaccharides, alkaloids, organic acids, volatile oils, and saponins) [1]. These compounds have the potential to address the issues of microbial resistance and drug residues associated with antibiotic use in the enhancement of animal production performance [1,2]. Tea has traditionally served as a health supplement, offering a myriad of benefits, including antioxidant, antibacterial, and anticancer properties and the regulation of lipid metabolism [3]. China holds the distinction of being the foremost producer and consumer of tea, having engaged in the cultivation and utilization of the tea plant for over 3000 years. The production and processing of tea yield various by−products, including dried tea, tea juice, tea residue, and tea powder, which are rich in bioactive compounds such as tea polyphenols [4,5]. Tea polyphenols (TPs) are polyhydroxy−phenolic compounds extracted from tea leaves which not only have antioxidant, antitumor, anti−aging and immunomodulatory properties but also regulate the composition and function of the intestinal microbiota and protect liver health [6,7,8,9]. This TP study establishes a theoretical framework for the development and utilization of tea processing by−products as a feed source to support animal health, thus making an important contribution to the sustainable development of the livestock industry. It is worth noting that TP can act on the “microbe–gut–brain” [9] axis and “gut–liver “axis [7]. However, individual differences in the composition of gut microbiota may also lead to differences in the bioavailability and bioefficacy of polyphenols and their metabolites [10]. Only a limited number of gut microbes have been identified as capable of catalyzing phenolic metabolism and its corresponding catabolic pathways, such as *Bifidobacterium* sp., *Lactobacillus* sp., *Bacteroides* sp., and *Eubacterium* sp. Furthermore, the situation becomes even more complex when considering “polyphenol–microbiota” interactions [11,12].

Several studies have demonstrated that the genera *Lachnospiraceae*, *Bacteroides*, *Alistipes*, and *Faecalibaculum* can act as biomarkers for the redox state of the intestine. It has been observed that varying doses of TPs can influence both the intestinal redox state and the composition of the intestinal microbiota in distinct ways. Moreover, excessive intake of TPs has been found to diminish the positive effects on intestinal health [13]. Furthermore, TPs demonstrated the ability to counteract the decrease in gut microbiota abundance and diversity caused by antibiotic treatment. This was achieved by notably enhancing the prevalence of beneficial microorganisms like *Lactobacillus*, *Akkermansia*, *Blautia*, *Roseburia,* and *Eubacterium* [14]. Meanwhile, TPs also reduced intestinal inflammation and oxidative stress−induced infections, decreased the abundance of *Bacteroides*, *Escherichiashigella*, *Anaerotruncus*, etc., and enhanced the expression of tight proteins to modulate *Salmonella typhimurium*−induced intestinal microbiota disorders in mice [15]. Interestingly, TPs also inhibit lipogenesis and promote fat oxidation by reducing inflammatory factors, enhancing liver metabolic regulation, and altering the composition of specific microbiota and the expression of relevant genes, and promote gluconeogenic pathways and regulate host energy metabolism [16,17]. TPs increase energy conversion by increasing beneficial microorganisms (such as *C. Ruminococcaceae*, *C. Lachnospiraceae* and *B. Bacteroidaceae*), promoting the mitochondrial TCA cycle and urea cycle of intestinal microorganisms, thereby accelerating fat consumption and decreasing fat accumulation [18]. It was found that TPs up−regulated liver Cu/Zn−SOD, Mn−SOD, GSH−Px, PPAR−α, LDLR, and CPT−1a gene expression and down−regulated COX−2, iNOS, IκB−α, FAS, SREBP1−c, and C/EBP−α gene expression to enhance liver antioxidant capacity and lipid metabolism regulation [7]. Tea demonstrated the ability to mitigate liver bile acid disorders induced by a high−fat diet and effectively normalized bile acid levels in mice subjected to various dietary conditions [19]. Furthermore, TPs mitigate disturbances in liver purine and amino acid metabolism caused by hyperlipidemia and are also involved in riboflavin, glycerophospholipid, pyrimidine, and arachidonic acid metabolism [20].

Ruminants exhibit intricate rumen fermentation processes that allow them to transform non−starchy plant−based carbohydrates, which are indigestible for humans, into consumable products like meat and milk [21]. These products serve as valuable sources of dietary proteins for human consumption. Consequently, the preservation of rumen health in ruminants holds significant importance. Nevertheless, the distinctive rumen functionality in ruminants significantly varies in metabolic physiology compared to that of humans, mice, and poultry. In view of the aforementioned numerous biological functions of tea polyphenols found in humans [3,12], mice [7,15], and poultry [2,22], and the potential value of replacing antibiotics in livestock production systems to maintain homeostasis, it was hypothesized that TP have the potential to enhance rumen metabolism and subsequently improve liver metabolism. Thus, the current study was conducted to investigate this hypothesis.

## 2. Materials and Methods

### 2.1. Ethics Statement

All experimental designs and feeding management involving animals were approved by the Institute of Animal Husbandry and Veterinary, Jiangxi Academy of Agricultural Sciences (2010–JAAS–XM–01). The experiment was conducted at the Science and Technology Service Workstation of Jiangxi Academy of Agricultural Sciences (Ganzhou Lvlinwan Agriculture and Animal Husbandry Co., Ltd., Ganzhou, China).

### 2.2. Animals, Diet, and Experiment Design

A total of 60 male Hu sheep (lambs) with similar initial body weights (15.19 ± 0.75 kg) at 3 months of age were chosen and divided randomly into four groups, each consisting of 15 sheep. The control group (CON group) received a basal diet, while the experimental groups (TP400, TP800, and TP1200 groups) were provided with 400, 800, and 1200 mg/kg of TPs, respectively, added to the basal diet over a 72−day experimental period. The amount of added tea polyphenols was based on the literature reports [23,24,25,26]. Basal diets were formulated according to the nutritional requirements of Hu sheep (2004), and the feed ingredients and nutrient composition of the diets are shown in Table 1. During the study period, Hu sheep were allowed to take feed and water ad libitum. Feeding management adhered to farm regulations, with the Hu sheep being maintained in a consistent feeding environment and receiving feed twice daily at 08:30 and 17:30.

### 2.3. Sample Collection

Six lambs of average weight were selected from each treatment group for serum indices, rumen−related indices, and liver metabolome analysis.

Before feeding at the end of the experiment, 5 mL of blood was collected from the jugular vein using a blood collection tube (Jiangxi Jingzhi Technology Co., Ltd., Nanchang, China), centrifuged (3500 rpm, 10 min) in a low−speed centrifuge (TDL–80–2B; Anting Scientific Instrument Factory, Shanghai, China), and stored at −80 °C for serum indicator analysis. Serum indicators were determined according to the kit instructions (Shanghai Enzyme linked Biotechnology Co., Ltd., Shanghai, China), including low−density lipoprotein (LDL), high−density lipoprotein (HDL), glucose (GLU), urea nitrogen (UN), total cholesterol (TC), triglyceride (TG), immunoglobulins (IgG, IgM, and IgA), glutathione peroxidase (GSH−Px), catalase (CAT), total superoxide dismutase (TSOD), and malondialdehyde (MDA).

Referring to the method of Wang et al. [27], 50 mL of rumen fluid was collected from four treatment groups of Hu sheep using a gastric tube rumen sampler prior to feeding, of which three portions were sent to the laboratory via liquid nitrogen in a freezing tube to be preserved at −80 °C for subsequent analysis. Moreover, a portable pH meter (PHBJ−260F; INESA Scientific Instruments Co., LTD, Shanghai, China) was used to immediately determine rumen fluid pH. The rumen fluid NH_3_−N content was determined with reference to the method of Li [28] et al. The VFA molar content, including acetic acid (AA), propionic acid (PA), butyric acid (BA), and TVFAs, was determined using a gas chromatograph (GC−7890B, Agilent Technologies, Petaling Jaya, Malaysia) with reference to the method described in detail by Wang et al. [27]. The VFA molar ratios and acetic acid/propionic acid (AA/PA) were calculated. Six lambs of average weight per group were selected for slaughter. Samples from the same part of the liver were immediately collected in freezing tubes, then snap−frozen in liquid nitrogen and transported to the laboratory for storage at −80 °C until liver metabolome analysis.

### 2.4. DNA Extraction and Analysis of Bacterial Community in Rumen

The bacterial DNA was extracted from 24 rumen samples using the TGuide S96 Magnetic Stool DNA Kit (Tiangen Biotech (Beijing, China) Co., Ltd., Beijing, China) according to manufacturer instructions. The DNA concentration of the samples was measured with the Qubit dsDNA HS Assay Kit and Qubit 4.0 Fluorometer (Invitrogen, Thermo Fisher Scientific, OR, Eugene, USA), which was qualified and used for the amplification of full−length 16S rRNA gene amplification (27F: AGRGTTTGATYNTGGCTCAG and 1492R: TASGGHTACCTTGTTASGACTT). In the meantime, both the forward and reverse 16S primers were tailed with sample−specific PacBio barcode sequences to allow for multiplexed sequencing. The KOD One PCR Master Mix (TOYOBOLife Science, Osaka, Japan) was used to perform 25 cycles of PCR amplification, with initial denaturation at 95 °C × 2 min, followed by 25 cycles of denaturation at 98 °C × 10 s, 55 °C × 30 s, 72 °C × 90 s, and then 72 °C × 2 min. PCR products were purified (Agencourt AMPure XP Beads, Beckman Coulter, Indianapolis, IN, USA), quantified (Qubit dsDNA HS Assay Kit and Qubit 4.0 Fluorometer, Invitrogen, Thermo Fisher Scientific, Oregon, USA), and normalized to form a sequencing library (SMRT Bell, Pacific Biosciences, Menlo Park, CA, USA). Purified SMRTbell libraries from the pooled and barcoded samples were sequenced on a single PacBio Sequel II 8M cell using the Sequel II Sequencing kit 2.0. Sequencing was performed by Beijing Biomarker Technologies Co., Ltd. (Beijing, China). First, we exported CCS (Circular Consensus Sequencing) files from the raw reads generated from sequencing using SMRT Link software v13.0. Subsequently, the CCS was identified by barcode to obtain Raw−CCS sequence data, followed by cutadapt (version 2.7) software to identify and filter to obtain Clean−CCS sequences. After that, Effective−CCS sequences were obtained by identifying and removing chimeric sequences using the UCHIME algorithm (version 8.1) software, which clustered the sequences using USEARCH (v10.0) [29]. Finally, the classification was labeled using the SILVA database (release 132) [30] and the naive Bayes classifier in QIIME2 [31] at a 70% confidence threshold.

### 2.5. Metabolome Sequencing and Bioinformatics Analysis

Metabolite assays are described in detail with reference to the previous article by Wang et al. [32]. Briefly, 50 mg of liver in 1000 μL (100 μL of rumen fluid taken in 500 μL) of extraction solution containing internal standard was vortexed and mixed (30 s). Next, magnetic beads were added for grinding, subjected to sonication, and the supernatant was taken for vacuum drying after standing and centrifugation. Subsequently, an appropriate amount of the extract was added for reconstitution, and the raw data were obtained using Waters Acquity I−Class PLUS ultra−high performance liquid tandem Waters Xevo G2−XS QT high−resolution mass spectrometer (Waters, Milford, MA, USA) under the control of acquisition software (MassLynx V4.2, Waters, Milford, CT, USA). Finally, metabolite identification was performed by Progenesis QI (version 4.0) software [33]. Finally, the identified metabolites were annotated using the KEGG (https://www.genome.jp/kegg/, accessed on 23 July 2024) database, and the annotated metabolites were then mapped to the KEGG Pathway database [34,35].

### 2.6. Statistical Analysis

It was verified that the serum and rumen phenotypic data for the sheep conformed to normal distribution. Then, the serum physiology, rumen fluid pH, NH3−N, VFA molar concentration, and rumen VFA molar proportion data that conformed to normal distribution were subjected to one−way ANOVA using SPSS software. (version 26.0) (SPSS Inc., Chicago, IL, USA). Data were analyzed using linear and quadratic effects and multiple comparisons were performed using Duncan’s method when ANOVA showed significant differences. (*p* < 0.05). However, the Kruskal−Wallis test was performed for serum immune and antioxidant indices that did not conform to normal distribution and significance analyses were performed for multiple comparisons. The alpha diversity index (Chao1, Ace, Shannon, Simpson) of the sample was evaluated using QIIME2 (https://qiime2.org, accessed on 6 June 2020) software. The beta diversity assessment was based on binary Jaccard distance by principal coordinate analysis (PCoA) and non−metric multidimensional scaling (NMDS), where significant differences among the four treatment groups were evaluated using ANOSIM. LEfSe (linear discriminant analysis (LDA = 3) size) was used to identify statistically different biomarkers between groups [36]. The sequence was based on Kyoto Encyclopedia of Genes and Genomes (KEGG) annotation pathways in PICRUSt (v2.2.0−b) [37], with pathway significance analyzed using nonparametric tests. The orthogonal projections to latent structures discriminate analysis (OPLS−DA [38]) model identified Variable Importance in Projection (VIP) values for screening differential metabolites in each treatment group in the rumen or liver. Finally, identified metabolites were annotated using the KEGG database to construct liver or rumen top 5 differential metabolite enrichment network maps [34,35].

## 3. Results

### 3.1. Effect of Dietary Supplementation with TPs on Serum and Rumen Fermentation in Hu Sheep

#### 3.1.1. Effect of Dietary Supplementation with TPs on Serum Biochemistry, Immunity, and Antioxidants in Hu Sheep

Hu sheep serum biochemistry, serum immunity, and antioxidants were determined and are exhibited in Table 2 and Table 3. The results showed that TP supplementation in Hu sheep diets had no significant effect on serum biochemistry (LDL, HDL, GLU, UN, TC, and TG), serum immunity (IgG and IgM), and serum antioxidants (MDA) (*p* > 0.05). Compared to the CON group, TP supplementation in the diet significantly increased serum IgA, GSH−Px, and TSOD (*p* < 0.05). Furthermore, TP400, TP800, and TP1200 exhibited no significant changes in serum LDL, HDL, GLU, UN, TC, TG, IgG, IgM, TSOD, and MDA (*p* > 0.05). In addition, compared to the CON and TP400 groups, the TP800 groups significantly reduced CAT (*p* < 0.05). Compared to the TP400 groups, the TP1200 groups significantly increased IgA but significantly reduced CAT (*p* < 0.05).

CON: control group; TP400: the basal diet was supplemented with 400 mg/kg of tea polyphenols; TP800: the basal diet was supplemented with 800 mg/kg of tea polyphenols; TP1200: the basal diet was supplemented with 1200 mg/kg of tea polyphenols; SEM: standard error of mean; LDL: low−density lipoprotein; HDL: high−density lipoprotein; GLU: glucose; UN: urea nitrogen; TC: total cholesterol; TG: triglyceride.

CON: control group; TP400: the basal diet was supplemented with 400 mg/kg of tea polyphenols; TP800: the basal diet was supplemented with 800 mg/kg of tea polyphenols; TP1200: the basal diet was supplemented with 1200 mg/kg of tea polyphenols; SEM: standard error of mean; IgG: immunoglobulin G; IgM: immunoglobulin M; IgA: immunoglobulin A; GSH−Px: glutathione peroxidase; CAT: catalase; TSOD: total superoxide dismutase; and MDA: malondialdehyde.Peer numbers without shoulder letters or with the same letter indicate non−significant differences, whereas different letters indicate significant differences.

#### 3.1.2. Effect of Dietary Supplementation with TPs on Rumen Fermentation in Hu Sheep

The results showed in Table 4 that TP supplementation in the diet of Hu sheep had no significant effect on NH_3_−N, VFA molar concentration (AA, PA, BA, and TVFAs), AA/PA, and VFA molar proportion (AAR and PAR) in the rumen (*p* > 0.05). Compared to the CON group, TP supplementation in the diet significantly increased rumen fluid pH (*p* < 0.05). Meanwhile, the rumen fluid BAR of TP400 was significantly higher than that of the CON and T800 groups (*p* < 0.05). Furthermore, the TP400, TP800, and TP1200 groups exhibited no significant changes in rumen pH, NH_3_−N, AA, PA, BA, TVFAs, AAR, and PAR (*p* > 0.05). In addition, compared to the TP400 group, the TP800 group had significantly reduced BAR (*p* < 0.05).

### 3.2. Effect of Feed Supplementation with Tea Polyphenols on Rumen Bacteria in Hu Sheep

#### 3.2.1. Analysis of Rumen Microbial Diversity

The rumen microorganism alpha and beta diversity estimation of the CON, TP200, TP400, TP800, and TP1200 groups are shown in Table 5 and Figure 1, respectively. Analyzing Table 5, it can be found that feed supplementation with TPs had no significant effect on the alpha diversity ACE (*p* = 0.884), Chao1 (*p* = 0.906), Simpson (*p* = 0.411), and Shannon (*p* = 0.644) of rumen microorganisms in Hu sheep. PCoA results showed significant separation between the TP400, TP800, and TP1200 groups and the CON group (Figure 1A). The NMDS analysis shows a stress value of 0.1958 < 0.2, indicating model reliability (Figure 1B). The ANOSIM analysis reveals R = 0.575 and *p* = 0.001, indicating significant group differences (Figure 1B). In conclusion, the supplementation of TPs to the diet was able to affect the rumen microbial beta diversity.

#### 3.2.2. Effect of Feed Supplementation with TPs on the Composition and Function of Rumen Microorganisms

In the microbial composition analysis, we focused on the top 10 microbial levels at the phylum (Figure 2A) and genus (Figure 2B) levels. At the phylum level, Firmicutes is the most abundant, accounting for 64.85%, 66.11%, 61.90%, and 68.60% in the CON, TP400, TP800, and TP1200 lamb groups (*p* = 0.778), respectively (Figure 2A). Bacteroidetes (CON: 23.30%, TP400: 22.43%, TP800: 28.08%, and TP800: 21.31%; *p* = 0.741) and Kiritimatiellaeota (CON: 4.57%, TP400: 6.12%, TP800: 3.22%, and TP800: 1.78%; *p* = 0.084) were, respectively, the second and third most abundant phyla based on 16S rRNA sequencing. At the genus level, in the CON group, we found the top four dominant bacteria as *Succiniclasticum* (*p* = 0.312), *Rikenellaceae_RC9_gut_group* (*p* = 0.311), *uncultured_bacterium_f_Veillonellaceae* (*p* = 0.529), and *Prevotella_1* (*p* = 0.333). However, in the TP400 and TP1200 groups, the top four were *Succiniclasticum* (*p* = 0.312), *Christensenellaceae_R*−*7_group* (*p* = 0.115), *Rikenellaceae_RC9_gut_group* (*p* = 0.311), and *uncultured_bacterium_f_Veillonellaceae* (*p* = 0.529), but in the TP800 group, the top four were *Rikenellaceae_RC9_gut_group* (*p* = 0.311), *Succiniclasticum* (*p* = 0.312), *Quinella* (*p* = 0.177), and *Prevotella_1* (*p* = 0.333) (Figure 2B). However, there was no significant effect of the top 10 microbial levels at both the phylum and genus levels.

Further analysis by Venn diagram revealed the presence of 12 (OTU1071, OTU1078, OTU1090, OTU1195, OTU332, OTU377, OTU401, OTU5, OTU516, OTU521, OTU548, OTU991), 13 (OTU1249, OTU1257, OTU1260, OTU1280, OTU1282, OTU1284, OTU171, OTU287, OTU31, OTU478, OTU904, OTU906, OTU912), 8 (OTU1333, OTU1334, OTU1335, OTU1345, OTU161, OTU184, OTU324, OTU731), and 8 (OTU1235, OTU15, OTU312, OTU318, OTU358, OTU50, OTU555, OTU569) OTUs in the CON, TP400, TP800, and TP1200 groups, respectively (Figure 2C). Subsequently, the effects of supplemental TPs on rumen biomarkers in lambs were analyzed using LEfSe with an LDA of 3.0. The analysis found 37 biomarkers in lamb rumen fluid across treatment groups: 11 for CON, 7 for TP400, 1 for TP800, and 18 for TP1200 (Figure 2D). At the genus level, the CON group was included in *Selenomonas_1*, *Suttonella*, *[Eubacterium]_ruminantium_group*, and *Candidatus_Saccharimonas*. The three biomarkers in the TP400 group were *uncultured_bacterium_o_WCHB1*−*41*, *Butyrivibrio*, and *Eubacterium*. The one biomarker in the TP800 group was *Anaerovibrio.* The five biomarkers in the TP1200 group were *uncultured_bacterium_o_Mollicutes_RF39*, *Fretibacterium*, *Ruminococcaceae_UCG*−*014*, *Lachnospiraceae_AC2044_group*, *Coprococcus_1*, and *uncultured_bacterium_k_Bacteria* (Figure 2D).

We predicted rumen microbial function in the CON, TP400, TP800, and TP1200 groups using PICRUSt2 and the KEGG database. Results showed that metabolic pathways were the main enrichment, accounting for over 71% in each group, including global and overview maps, carbohydrate metabolism, amino acid metabolism, metabolism of cofactors and vitamins, energy metabolism, and nucleotide metabolism (Figure 2E). Furthermore, adding TPs to the diet had no significant effect on rumen microbial function in Class−2−level top 10 pathways (Figure 2E). Finally, a network diagram of genus−level communities at the top 80% of abundance showed that Firmicutes dominated, with the strongest correlation between the *Ruminococcaceae_NK4A214_group* and *Christensenellaceae_R*−*7_group* (*p* = 0.000) (Figure 2F).

### 3.3. Effect of Feed Supplementation with TPs on Rumen Metabolism in Hu Sheep

#### 3.3.1. Analysis of Rumen Differential Metabolites across the Four Groups

It can be seen from OPLS−DA that the samples in each group were distinguished. In this case, the model is reliable (Q2Y > 0.50) and can be used to screen for differential metabolites (Figure 3A−F). Subsequently, select screening for differential metabolites between comparison groups was performed using VIP > 1 and *p* < 0.05 as a threshold. Meanwhile, we constructed differential metabolite number histograms and found that there were 154, 313, 179, 320, 222, and 262 differential metabolites distributed in CON vs. TP400 (92 up−regulated and 62 down−regulated, *p* < 0.05), CON vs. TP800 (211 up−regulated and 102 down−regulated, *p* < 0.05), CON vs. TP1200 (99 up−regulated and 80 down−regulated, *p* < 0.05), TP400 vs. TP800 (180 up−regulated and 140 down−regulated, *p* < 0.05), TP400 vs. TP1200 (95 up−regulated and 127 down−regulated, *p* < 0.05), and TP800 vs. TP1200 (94 up−regulated and 168 down−regulated, *p* < 0.05) (Figure 3G). Further, we constructed a differential metabolite Venn diagram and found that there were 19, 47, 20, 47, 27, and 31 unique differential metabolites distributed in CON vs. TP400, CON vs. TP800, CON vs. TP1200, TP400 vs. TP800, TP400 vs. TP1200, and TP800 vs. TP1200, respectively (Figure 3H).

#### 3.3.2. Rumen Functional Enrichment Analysis of Differential Metabolites

Figure 4 demonstrates KEGG enriched point maps and top five enriched network maps of rumen metabolites. First, the CON vs. TP400 differential metabolite enrichment maps and top five enrichment network maps were analyzed and are presented in Figure 4A,B, which are mainly enriched in the biosynthesis of plant hormones (*p* = 0.007), biosynthesis of various alkaloids (*p* = 0.015), glucosinolate biosynthesis (*p* = 0.015), inflammatory mediator regulation of TRP channel lysine degradation (*p* = 0.009), and phenylalanine, tyrosine, and tryptophan (*p* = 0.001). Next, the CON vs. TP800 differential metabolite enrichment maps and top five enrichment network maps were analyzed and are presented in Figure 4C,D, which are mainly enriched in the biosynthesis of alkaloids derived from histidine and purine (*p* = 0.030), carbon fixation pathways in prokaryotes (*p* = 0.012), central carbon metabolism in cancer (*p* = 0.041), citrate cycle (TCA cycle) (*p* = 0.043), and purine metabolism (*p* = 0.014). Subsequently, the CON vs. TP1200 differential metabolite enrichment maps and top five enrichment network maps were analyzed and are presented in Figure 4E,F, which are mainly enriched in the carbon fixation pathways in prokaryotes (*p* = 0.017), diterpenoid biosynthesis (*p* = 0.045), galactose metabolism (*p* = 0.045), lysine degradation (*p* = 0.011), and cAMP signaling pathway (*p* = 0.045).

Moreover, we further analyzed the differential metabolite enrichment maps and top five enrichment network maps for different TP doses. First, the TP400 vs. TP800 differential metabolite enrichment maps and top five enrichment network maps were analyzed and are presented in Figure 4G,H, which are mainly enriched in the biosynthesis of alkaloids derived from histidine and purine (*p* = 0.006), central carbon metabolism in cancer (*p* = 0.019), sulfur metabolism (*p* = 0.019), teichoic acid biosynthesis (*p* = 0.019), and cAMP signaling pathway (*p* = 0.019). Next, the TP400 vs. TP1200 differential metabolite enrichment maps and top five enrichment network maps were analyzed and are presented in Figure 4I,J, which are mainly enriched in the biosynthesis of various alkaloids (*p* = 0.030), glucosinolate biosynthesis (*p* = 0.030), propanoate metabolism (*p* = 0.035), sulfur metabolism (*p* = 0.005), and sulfur relay system (*p* = 0.028). Finally, the TP800 vs. TP1200 differential metabolite enrichment maps and top five enrichment network maps were analyzed and are presented in Figure 4K,L, which are mainly enriched in choline metabolism in cancer (*p* = 0.025); neomycin, kanamycin, and gentamicin biosynthesis (*p* = 0.038); taste transduction (*p* = 0.006); teichoic acid biosynthesis (*p* = 0.007); and the two−component system (*p* = 0.025).

#### 3.3.3. Analysis of the Correlation between Rumen Differential Metabolites and Liver Metabolism, Rumen Microbiology, and Phenotype

Differential metabolites in the rumen between treatment groups were screened based on VIP > 1.9 and AUC = 1, and correlation analysis of differential metabolites with liver metabolism, rumen and serum phenotypes, and rumen microbes was performed, and the results are shown in Figure 5A. As can be seen from their correlation coefficients obtained between rumen differential metabolites, most of the correlation coefficients were mostly less than 0.5 in absolute value. However, there were some significant correlations between them between some of the differential metabolites. Mantel’s r analysis revealed correlations between liver metabolism, rumen and serum phenotypes, and rumen microbes with rumen differential metabolites, with the most significant correlation with neg_4785 (LysoPE (18_0_0_0), liver metabolism (*p* = 0.075), rumen and serum phenotypes (*p* = 0.562), rumen microbe phylum levels (*p* = 0.018) and genus levels (*p* = 0.024)). In order, the top five were neg_4785 (LysoPE(18_0_0_0)), neg_5476 (estradiol valerate, liver metabolism (*p* = 0.351), rumen and serum phenotypes (*p* = 0.001), rumen microbe phylum levels (*p* = 0.865) and genus levels (*p* = 0.764)), pos_3975 (biliverdin, liver metabolism (*p* = 0.301), rumen and serum phenotypes (*p* = 0.004), rumen microbe phylum levels (*p* = 0.989) and genus levels (*p* = 0.974)), pos_1420 (10−Hydroxydesmethylnortriptyline glucuronide, liver metabolism (*p* = 0.030), rumen and serum phenotypes (*p* = 0.612), rumen microbe phylum levels (*p* = 0.084) and genus levels (*p* = 0.083)), and pos_7497 (Kinetin, liver metabolism (*p* = 0.026), rumen and serum phenotypes (*p* = 0.959), rumen microbe phylum levels (*p* = 0.168) and genus levels (*p* = 0.130)) (Figure 5A). Further correlation analysis between rumen microbial markers and KEGG enrichment network top five metabolites revealed that there were essentially no significant differences between rumen markers and KEGG enrichment network map top five metabolites. In addition, the absolute value of their correlation coefficients was overall less than 0.5, with the strongest correlation between neg_1411 (Xanthine) and *Eubacterium* (r = 0.686, *p* = 0.000) (Figure 5B). In addition, rumen differential markers *Coprococcus_1*, *Lachnospiraceae_AC2044_group*, *uncultured_bacterium_k_Bacteria*, *Candidatus_Saccharimonas*, *Ruminococcaceae_UCG*−*014*, and *[Eubacterium]_ruminantium_group* had essentially no significant effect on rumen KEGG top five enriched metabolites (*p* > 0.05) (Figure 5B). However, *uncultured_bacterium_o_WCHB1*−*41*, *Eubacterium*, *Butyrivibrio*, *Fretibacterium*, *uncultured_bacterium_o_Mollicutes_RF39*, and *Anaerovibrio* significantly affected the KEGG top five differential metabolites (*p* < 0.05) (Figure 5B).

### 3.4. Effect of Feed Supplementation with TPs on Liver Metabolism in Hu Sheep

#### 3.4.1. Analysis of Liver Differential Metabolites across the Four Groups

It can be seen from the OPLS−DA that the samples of each group were distinguished (Figure 6A–F). Subsequently, select screening for differential metabolites between comparison groups was performed using VIP > 1 and *p* < 0.05 as a threshold. Meanwhile, we constructed differential metabolite histograms and found that there were 122, 209, 165, 145, 95, and 94 differential metabolites distributed in CON vs. TP400 (69 up−regulated and 53 down−regulated, *p* < 0.05), CON vs. TP800 (52 up−regulated and 157 down−regulated, *p* < 0.05), CON vs. TP1200 (73 up−regulated and 92 down−regulated, *p* < 0.05), TP400 vs. TP800 (30 up−regulated and 115 down−regulated, *p* < 0.05), TP400 vs. TP1200 (49 up−regulated and 46 down−regulated, *p* < 0.05), and TP800 vs. TP1200 (74 up−regulated and 20 down−regulated, *p* < 0.05) (Figure 6G). Further, we constructed a differential metabolite Venn diagram and found that there were 28, 59, 46, 32, 15, and 19 unique differential metabolites distributed in CON vs. TP400, CON vs. TP800, CON vs. TP1200, TP400 vs. TP800, TP400 vs. TP1200, and TP800 vs. TP1200, respectively (Figure 6H).

#### 3.4.2. Functional Enrichment Analysis of Differential Metabolites in Liver

Liver metabolites were annotated with the KEGG database, and Figure 7A shows the top 20 pathways with the most selected metabolites, including amino acid metabolism (3 pathways, including tyrosine, cysteine and methionine, and phenylalanine), digestive system (2 pathways, including bile secretion, and protein digestion and absorption), cancer: overview (1 pathway, central carbon metabolism in cancer), biosynthesis of other secondary metabolites (1 pathway, biosynthesis of various plant secondary metabolites), lipid metabolism (3 pathways, including arachidonic acid, steroid hormone biosynthesis, and biosynthesis of unsaturated fatty acids), chemical structure transformation maps (5 pathways, including biosynthesis of plant secondary metabolites; biosynthesis of phenylpropanoids; biosynthesis of plant hormones; biosynthesis of alkaloids derived from ornithine, lysine, and nicotinic acid; and biosynthesis of alkaloids derived from the shikimate pathway), membrane transport (1 pathway, ABC transporters), metabolism of other amino acids (1 pathway, D−amino acid metabolism), nucleotide metabolism (1 pathway, pyrimidine metabolism), signaling molecules and interactions (1 pathway, neuroactive ligand–receptor interactions), and translation (1 pathway, Aminoacyl−tRNA biosynthesis).

Moreover, the network diagram of KEGG top five differential metabolite enrichment of the liver was constructed and is shown in Figure 7. First, the top five pathways enriched for CON vs. TP400 differential metabolites were analyzed, as shown in Figure 7B, which were mainly enriched in inflammatory mediator regulation of TRP channels (*p* = 0.057), pyrimidine metabolism (*p* = 0.038), steroid biosynthesis (*p* = 0.009), tyrosine metabolism (*p* = 0.038), and vascular smooth muscle contraction (*p* = 0.057). Next, the top five pathway analysis of CON vs. TP800 differential metabolite enrichment is shown in Figure 7C, which was mainly enriched in arachidonic acid metabolism (*p* = 0.031), inflammatory mediator regulation of TRP channels (*p* = 0.086), linoleic acid metabolism (*p* = 0.018), neuroactive ligand–receptor interactions (*p* = 0.031), and terpenoid backbone biosynthesis (*p* = 0.132). Subsequently, the top five pathway analysis of CON vs. TP1200 differential metabolite enrichment is shown in Figure 7D, which was mainly enriched in cortisol synthesis and secretion (*p* = 0.038), Cushing syndrome (*p* = 0.038), fatty acid degradation (*p* = 0.038), insect hormone biosynthesis (*p* = 0.038), and primary bile acid biosynthesis (*p* = 0.057).

We further analyzed the effect of different doses of TPs on the enrichment of liver differential metabolites. First, the top five pathways analyzing the enrichment of TP400 vs. TP800 differential metabolites, shown in Figure 7E, were mainly enriched in arachidonic acid metabolism (*p* = 0.005), linoleic acid metabolism (*p* = 0.004), monoterpenoid biosynthesis (*p* = 0.017), platelet activation (*p* = 0.006), and vascular smooth muscle contraction (*p* = 0.002). Next, the top five pathways analyzed for TP400 vs. TP1200 differential metabolite enrichment, shown in Figure 7F, were mainly enriched in fatty acid elongation (*p* = 0.053); inflammatory mediator regulation of TRP channels (*p* = 0.015); PPAR signaling pathway (*p* = 0.053); parathyroid hormone synthesis, secretion, and action (*p* = 0.053); and secondary bile acid biosynthesis (*p* = 0.047). Finally, the top five pathways analyzed for TP800 vs. TP1200 differential metabolite enrichment, shown in Figure 7G, were mainly enriched in aldosterone synthesis and secretion (*p* = 0.048), biosynthesis of unsaturated fatty acids (*p* = 0.033), fatty acid biosynthesis (*p* = 0.030), linoleic acid metabolism (*p* = 0.048), and ubiquinone and other terpenoid–quinone biosynthesis (*p* = 0.011).

#### 3.4.3. Analysis of the Correlation between Liver Differential Metabolites and Rumen Metabolism, Rumen Microbiology, and Phenotype

Differential metabolites in the liver between treatment groups were screened based on VIP > 1.9 and AUC = 1, and correlation analysis of differential metabolites with rumen metabolism, rumen and serum phenotypes, and rumen microbes was performed, and the results are shown in Figure 8A. Overall, there were largely significant correlations between the liver differentiated metabolites obtained from the screening. Mantel’s r analysis revealed correlations between rumen metabolism, rumen and serum phenotypes, and rumen microbes with liver differential metabolites, with the most significant correlation with neg_3088 (alpha−D− (+) −mannose−1−phosphate, rumen metabolism (*p* = 0.695), rumen and serum phenotypes (*p* = 0.079), and rumen microbe phylum levels (*p* = 0.039) and genus levels (*p* = 0.014)). In order, the top five were neg_3088 (alpha−D− (+)−mannose−1−phosphate), neg_1050 (Valyl−Glutamine, rumen metabolism (*p* = 0.023), rumen and serum phenotypes (*p* = 0.015), rumen microbe phylum levels (*p* = 0.946) and genus levels (*p* = 0.949)), neg_2588 (ascorbic acid 6−palmitate, rumen metabolism (*p* = 0.001), rumen and serum phenotypes (*p* = 0.061), rumen microbe phylum levels (*p* = 0.986) and genus levels (*p* = 0.982)), pos_11852 (Glycocholic acid hydrate rumen metabolism (*p* = 0.014), rumen and serum phenotypes (*p* = 0.018), rumen microbe phylum levels (*p* = 0.737) and genus levels (*p* = 0.838)), and pos_12343 ((−)−Shinpterocarpin rumen metabolism (*p* = 0.001), rumen and serum phenotypes (*p* = 0.029), rumen microbe phylum levels (*p* = 0.849) and genus levels (*p* = 0.815)) (Figure 8A).

Further correlation analysis between rumen microbial markers and KEGG enrichment network top five metabolites revealed that the absolute value of their correlation coefficients was overall less than 0.5, with the strongest correlation between neg_5282 (Arachidonic acid) and *Anaerovibrio* (r = −0.776, *p* = 0.000) (Figure 8B). In addition, liver differential markers *Fretibacterium*, *Suttonella*, *uncultured_bacterium_o_Mollicutes_RF39*, *Ruminococcaceae_UCG*−*014*, *Candidatus_Saccharimonas*, and *Eubacterium* had essentially no significant effect on rumen KEGG top five enriched metabolites (*p* > 0.05) (Figure 8B). However, *Anaerovibrio*, *uncultured_bacterium_o_WCHB1*−*41*, *Selenomonas_1*, *Lachnospiraceae_AC2044_group*, *[Eubacterium]_ruminantium_group*, *Butyrivibrio*, *Coprococcus_1*, and *uncultured_bacterium_k_Bacteria* significantly affected the KEGG top five differential metabolites (*p* < 0.05) (Figure 8B).

### 3.5. Correlation Analysis of Rumen and Liver Differential Metabolites

The heat map between liver and rumen metabolites based on a correlation coefficient threshold of 0.8 is shown in Figure 9. Analysis of Figure 9 shows that there is significant correlation between liver and rumen metabolites, with rumen succinic acid showing the strongest positive correlation with liver alanyl−serine and methylmalonic acid. Among them, the rumen metabolites neg_1876, neg_1792, neg_1098, neg_2455, pos_3258, and pos_796 showed negative correlation with the liver exclusion of both pos_3820 and neg_2588 metabolites (*p* < 0.05). Moreover, rumen metabolites pos_6911, pos_7310, pos_5414, pos_2527, pos_11800, neg_1443, neg_1387, pos_5926, pos_9784, and pos_823 showed positive correlations with metabolites from the liver (*p* < 0.05), but metabolites pos_3820, neg_ 2588, pos_11817, and pos_11683 were excluded. In addition, pos_11817 and pos_11683 were basically not significantly different from rumen metabolites, and rumen pos_207 was basically not significantly different from liver metabolites (*p* > 0.05).

### 3.6. Microbiome and Metabolome in Relation to Serum Indices and Rumen Parameters in Hu Sheep

Correlation analysis of the relationship between the microbiome–metabolome and serum indicators and rumen parameters in Hu sheep is presented in Figure 10A,B. Overall, there was a significant positive correlation between rumen VFA molar content (AA, PA, BA, TVFAs), VFA molar ratios (AAR and BAR), and AA/PA, but there was an overall significant negative correlation between PAR and other rumen parameters (Figure 10A). In addition, NH_3_−N was significantly positively correlated with rumen VFA molar content (AA, PA, BA, TVFAs) (Figure 10A). However, there was no overall significant correlation between serum immunity, serum antioxidants, and serum physiological indices (Figure 10A). It is noteworthy that serum UN was closely correlated with rumen parameters, showing significant positive correlations with rumen VFA molar content (AA, PA, BA, TVFAs), AA/PA, AAR, and NH_3_−N, but it was significantly negatively correlated with PAR (Figure 10A). Moreover, the microbiome and metabolome were significantly correlated with NH_3_−N, pH, IgG, IgM, IgA, GSH−Px, CAT, and HDL (Figure 10A).

Subsequently, we further analyzed the relationship between rumen microbial markers and rumen and serum phenotypes (Figure 10B). It was found that rumen VFA molar content (AA, PA, BA, TVFAs) was significantly and positively correlated with *Butyrivibrio* and *Coprococcus_1*, while it was significantly and negatively correlated with *Anaerovibrio* (Figure 10B). In addition, rumen BAR was significantly positively correlated with *Butyrivibrio* and Eubacterium, but it was significantly negatively correlated with *[Eubacterium]_ruminantium_group*, *uncultured_bacterium_o_Mollicutes_RF39*, and *Anaerovibrio* (Figure 10B). Subsequently, correlation analysis of rumen markers with serum immunoglobulins showed that *Fretibacterium* and *Anaerovibrio* were significantly negatively correlated with IgM. In addition, IgA was positively correlated with *Fretibacterium*, but it was significantly negatively correlated with *uncultured_bacterium_o_WCHB1*−*41* and *Selenomonas_1* (Figure 10B). Next, correlation analysis of rumen markers with serum antioxidant indices showed that *Suttonella* and *Selenomonas_1* were significantly negatively correlated with GSH−Px and TSOD. In addition, *uncultured_bacterium_o_Mollicutes_RF39* was significantly negatively correlated with CAT. However, *Eubacterium* and GSH−Px, *Fretibacterium* and TSOD, and *Candidatus_Saccharimonas* and MDA showed significant positive correlations (Figure 10B). Finally, correlation analysis of rumen markers with serum biochemicals showed that TC was significantly positively correlated with *[Eubacterium]_ruminantium_group* and *Suttonella*. In addition, *Ruminococcaceae_UCG*−*014* and TG, *Butyrivibrio* and UN, *uncultured_bacterium_o_Mollicutes_RF39* and LDL, and *Fretibacterium* and LDL showed significant positive correlations. However, *[Eubacterium]_ruminantium_group* and UN, *Candidatus_Saccharimonas* and HDL, and *uncultured_bacterium_o_Mollicutes_RF39* and HDL showed significant negative correlation (Figure 10B).

## 4. Discussion

### 4.1. Effect of TPs on Serum Biochemistry, Immunity, and Antioxidant Fermentation in Hu Sheep

TPs have been demonstrated to exhibit a range of biological activities, including antioxidant properties, immune system enhancement, regulation of intestinal microbiota, liver protection, and overall body health maintenance [6,7,8,9,39]. Consequently, TPs are widely used as a nutraceutical and feed additive for animal health [3,39,40,41]. Serum is the main carrier of nutrients and metabolic waste in the organism. It is essential for assessing health conditions and risks, reflecting overall health status, including immune, antioxidant, metabolic, and other functions. In the current experimental setup, the supplementation of 400, 800, and 1200 mg/kg TPs in the diet did not yield a significant impact on serum LDL, HDL, GLU, UN, TC, and TG levels in Hu sheep. This result is consistent with findings that TPs did not significantly affect serum biochemistry in dairy goats [42]. In conclusion, this dose of dietary TP supplementation does not negatively affect sheep.

Previous studies have shown that the branches and leaves trimmed from tea tree significantly increased the serum SOD and GSH−Px activity of Nanjiang yellow oats [43]. Moreover, TP can enhance the antioxidant capacity of the organism by increasing antioxidant enzymes in mice (Cu/Zn−SOD, Mn−SOD and GSH−Px) [7] and laying hens (T−AOC, SOD and GSH−Px) [22,39]. Additionally, TPs enhance intestinal antioxidant capacity (SOD and GSH−Px) and serum immunoglobulin (IgA and IgM) levels in weaned lambs [44]. Moreover, TPs may reduce oxidative stress and lipid peroxidation in hyperketonemic cows by boosting antioxidant enzyme activity (SOD and GPX) and lowering ROS and MDA levels [25]. The current study demonstrates that TP supplementation resulted in elevated levels of serum IgA, GSH−Px, and TSOD in fattening Hu sheep. This study demonstrated a close association between microbial markers and serum immunoglobulins and antioxidant indices. Additionally, liver and rumen differential metabolites had a significant impact on IgA, IgG, IgM, GSH−Px, and CAT levels. These findings are consistent with previous research indicating that TPs can enhance both the immune response and antioxidant capacity in animals [7,22,39,44]. This outcome may be associated with the ability of TPs to penetrate the animal organism and interact with reactive oxygen species (ROS), leading to the formation of relatively stable phenolic oxygen radicals, which subsequently neutralize free radicals or directly influence the regulation of antioxidant enzymes, including SOD and GSH−Px [3]. Furthermore, tea polyphenols may augment the body’s antioxidant capacity via the SOD pathway, which is mediated by the Nrf2/HO−1 [22,25,39,43,45]. Moreover, TPs can also modulate the NF−κB signaling pathway and mitigate the function of inflammatory damage [46]. Moreover, the potential boost in immune response and antioxidant capabilities in animals due to TPs may be attributed to the capacity of tea polyphenols to modulate microbiota and liver metabolism [8,20,47,48,49].

### 4.2. Effect of TPs on Rumen Microorganisms in Hu Sheep

Ruminant rumen microorganisms and their hosts co−evolve, generating small compounds through fermentation and forming a complex regulatory network [50,51]. In the present study, the addition of TPs to the diet did not significantly affect the rumen VFA molar content (AA, PA, BA, and TVFAs), AA/PA, AAR, BAR, and NH_3_−N. However, TP groups significantly increased the rumen pH, and the TP400 group significantly increased the BAR compared to the CON group. Rumen VFA molar content positively correlated with NH_3_−N, while both the VFA molar content and ratio negatively correlated with PAR. We hypothesized that the absence of significant changes in VFA molar content might be related to the absence of changes in the rumen microbiota. Previous studies indicate that microbial communities are influenced by genetics, sex, diet, age, environment, and geographic location, with observed similarities in rumen microbial structures [27,52,53,54,55]. This study found that adding TPs to the diet did not significantly affect the rumen microbiota at the phylum and genus levels of the top 10 of Hu sheep, with Firmicutes and Bacteroidetes remaining the dominant bacteria. This outcome supports earlier research showing that the primary rumen bacteria in sheep are Firmicutes and Bacteroidetes [27,56,57], which are essential for feed digestion, energy metabolism, and protection against pathogens [57,58,59]. Moreover, the network diagram of the top 80 genus−level communities showed that Firmicutes was dominant, with the strongest correlation between the *Ruminococcaceae_NK4A214_group* and *Christensenellaceae_R7_group*. In addition, the addition of TP to the diets of Hu sheep had no significant effect on the functional enrichment of rumen microbiota in the top 10. Previous studies on the effect of TPs on the microbiota of the rumen of ruminants are very limited. However, TPs were shown to regulate microbial community structure in the study of squabs [60], laying hens [2,61], mice [14,62,63], and humans [64,65]. This result may be related to the unique internal environment of the rumen in ruminants and requires further study.

The Mantel test is crucial for analyzing correlation studies between multiple categorical variables. The results of the Mantel’s r analysis showed no significant effect of rumen microbes on rumen AA, PA, BA, TVFAs, AA/PA, AAR, and PAR, which is consistent with the results of previous studies that found rumen microbes correlated with VFA in sheep [27]. We further analyzed rumen microbial genus level biomarkers, and VFA correlation showed that rumen VFA molar content (AA, PA, BA, TVFAs) was significantly and positively correlated with *Butyrivibrio* and *Coprococcus_1*, while it was significantly and negatively correlated with *Anaerovibrio*. Meanwhile, rumen BAR was significantly positively correlated with *Butyrivibrio* and Eubacterium, but it was significantly negatively correlated with *[Eubacterium]_ruminantium_group*, *uncultured_bacterium_o_Mollicutes_RF39*, and *Anaerovibrio*. This finding indicates that Mantel’s r primarily highlights linear associations within the matrix while disregarding nonlinear relationships. Moreover, it suggests the presence of intricate interactions among microorganisms that could potentially influence the correlations [52].

### 4.3. Effect of TPs on Rumen and Liver Metabolism in Hu Sheep

Metabolites bridge the gap between rumen microbes and host interactions. Analysis of metabolite KEGG enrichment revealed that the top five pathways for metabolite enrichment were chemical structure transformation maps, digestive system, amino acid metabolism, lipid metabolism, and membrane transport. First, rumen microbial markers were significantly associated with the top five differential metabolites in the KEGG enrichment network. Meanwhile, there were significant differences observed in the top five differential metabolite enrichment pathways of the KEGG between the rumen and liver across the TP400, TP800, and TP1200 groups. Furthermore, significant correlations existed between liver and rumen metabolites, with rumen succinic acid showing the strongest positive correlation with liver alanyl−serine and methylmalonic acid. Interestingly, succinic acid and methylmalonic acid are intermediates of the Krebs cycle, with succinic acid being particularly significant. Studies show that succinate is involved in inflammation, muscle protein synthesis, myofibril remodeling, energy supply, and glucose homeostasis [66,67].

Previous studies with the metabolome have revealed that amino acid metabolism, especially tryptophan metabolism, is critical in regulating muscle development and meat quality [68]. Furthermore, the TCA cycle plays an important role in the biosynthesis of macromolecules such as nucleotides, lipids, and proteins, and is involved in the control of transcription factors and chromatin modifications, which can alter cell function and fate [69]. Nevertheless, the molecular mechanisms by which variations in the levels of TCA cycle metabolites impact the expression of specific genes have yet to be fully elucidated in numerous instances. Concurrently, the cAMP signaling pathway serves as a vital cellular signaling pathway that governs various cellular processes, including metabolism (energy and lipid metabolism), growth, differentiation, and apoptosis [70,71]. In this study, the addition of TPs to diets was found to alter rumen and liver metabolite enrichment and was affected by dose. It was shown that with the dietary addition of TPs, rumen differential metabolites were mainly enriched in the biosynthesis of phenylalanine, tyrosine, and tryptophan; the TCA cycle; purine metabolism; galactose metabolism; lysine degradation; inflammatory mediator regulation of TRP channels; and the cAMP signaling pathway. Our further analysis of the enrichment of liver differential metabolites by dietary TP addition revealed significant differences between the liver differential metabolite enrichment pathway and the rumen. Liver differential metabolites were mainly enriched in the inflammatory mediator regulation of TRP channels, vascular smooth muscle contraction, cortisol synthesis and secretion, neuroactive ligand–receptor interactions, Cushing syndrome, and metabolism (pyrimidine metabolism, tyrosine metabolism, terpenoid backbone biosynthesis, insect hormone biosynthesis, steroid biosynthesis, arachidonic acid metabolism, linoleic acid metabolism, fatty acid degradation, and primary bile acid biosynthesis). The findings of this study are consistent with those of previous research, which indicated that TPs promote liver enrichment in both pyrimidine and arachidonic acid metabolism [20]. Furthermore, the study identified differences in the mechanisms by which TP regulates the liver and ruminal pathways. This result further suggests that TPs play an important function in the regulation of liver lipid metabolism [72,73,74]. Bile acids are composed of primary and secondary bile acids. Primary bile acids are produced by hepatocytes and stored in the gallbladder, whereas secondary bile acids are byproducts of bacterial metabolism. They play a crucial role in regulating cholesterol levels, eliminating endogenous and exogenous toxins, stimulating bile flow, and acting as signaling molecules that impact glucose homeostasis, lipid metabolism, and energy expenditure [75]. Primary bile acids are synthesized from cholesterol in the liver by either the classically or alternatively mediated pathways [76]. TPs may also affect liver metabolism by influencing gastrointestinal signaling pathways, suggesting a role in maintaining bodily homeostasis [7,77,78]. Research has indicated that TPs can lead to the down−regulation of genes involved in lipid anabolism, while there is a notable up−regulation in the expression levels of genes related to fat transportation and catabolism [41].

Mantel’s r analysis demonstrated correlations among liver metabolism, rumen and serum phenotypes, and rumen microbes in relation to differential metabolites in the rumen. The most significant correlation identified was with LysoPE (18_0_0_0). Moreover, rumen metabolism, rumen and serum phenotypes, and rumen microbial communities were identified in relation to liver differential metabolites, with the most pronounced correlation observed with alpha−D− (+)−mannose−1−phosphate. These results suggest a close correlation between rumen metabolism, liver metabolism, and rumen and serum phenotypes. However, Mantel’s r can only respond to linear relationships within the detection matrix [52]. Therefore, we speculate that they may still be closely related when Mantel’s r analysis is weakly correlated.

## 5. Conclusions

The results indicated that TPs did not significantly affect serum biochemistry or rumen bacteria composition, with Firmicutes dominating the network map of the top 80 genera and identifying 13 biomarkers at the genus level. Moreover, dietary supplementing with 400–800 mg/kg of TPs increases serum immunoglobulin and antioxidant capacity. In addition, TPs significantly altered rumen and liver differential metabolite enrichment pathways, among which rumen differential metabolites are mainly in metabolic biosynthesis; phenylalanine, tyrosine, and tryptophan; the TCA cycle; purine metabolism; galactose metabolism; lysine degradation; inflammatory mediator regulation of TRP channels; and the cAMP signaling pathway. However, they are enriched for liver differential metabolites, mainly in inflammatory mediator regulation of TRP channels, vascular smooth muscle contraction, cortisol synthesis and secretion, neuroactive ligand–receptor interactions, Cushing syndrome, fatty acid degradation, biosynthesis, and the pyrimidine, tyrosine, arachidonic acid, and linoleic acid metabolism pathways. Furthermore, a strong correlation exists between liver and rumen metabolites, especially rumen succinic acid and liver alanyl−serine and methylmalonic acid, which play crucial functions in enhancing immunity, antioxidants, and maintenance of body homeostasis. In conclusion, the best results were achieved with 400 mg/kg of TPs in the diets of fattening Hu sheep. This result established a theoretical foundation for TP application and future dosage optimization in sheep health.

## Figures and Tables

**Figure 1 animals-14-02661-f001:**
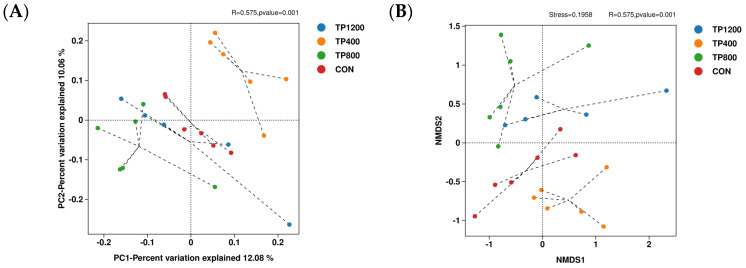
Beta diversity composition comparison of rumen microbiota in the CON, TP400, TP800, and TP1200 groups. (**A**) Principal coordinate analysis (PCoA) between treatment groups. (**B**) Non−metric multidimensional scaling (NMDS) analysis between treatment groups.

**Figure 2 animals-14-02661-f002:**
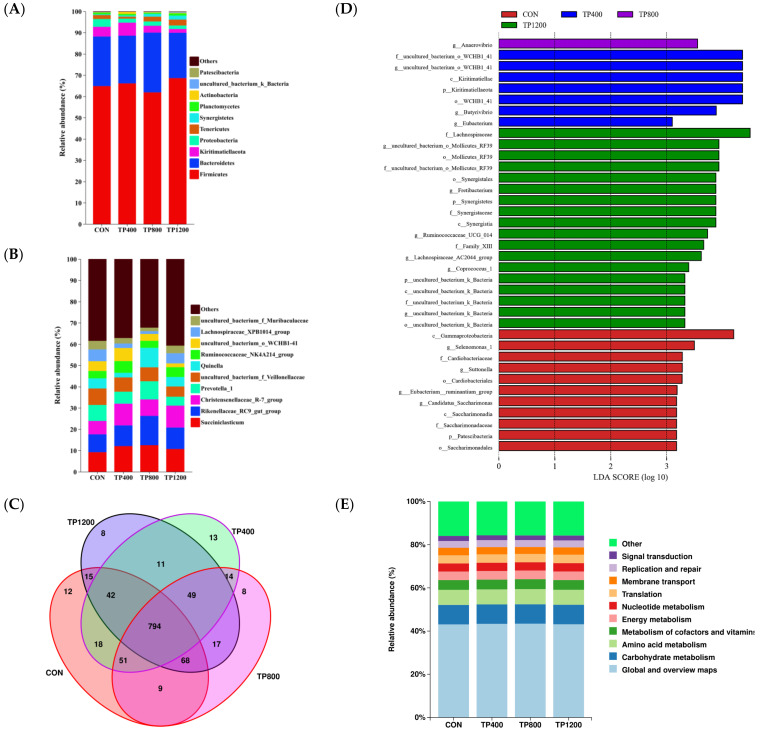
Composition and functional analysis of rumen microorganisms. Stacked histograms of the relative abundance of the rumen microbiota of the CON, TP200, TP400, TP800, and TP1200 groups at the phylum (**A**) and genus (**B**) levels of the top 10. (**C**) Venn diagram showing OTUs unique to and shared by each group. (**D**) Linear discriminant analysis effect size (LEfSe) analysis. (**E**) Microbial functional enrichment stacking diagram. (**F**) Correlation network maps were created for the top 80 abundant microorganisms at the genus level (*p* < 0.05). Among them, network diagram dot colors represent phyla, sizes indicate abundance, and line colors show correlation type—red for positive and green for negative—with thickness reflecting correlation size.

**Figure 3 animals-14-02661-f003:**
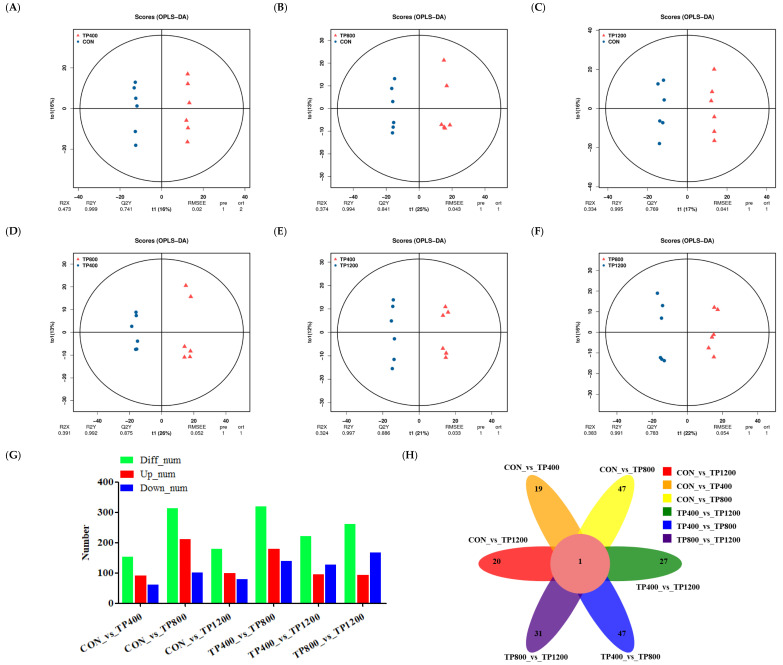
Analysis of rumen metabolite differences among the four groups. OPLS−DA ((**A**) CON vs. TP400, (**B**) CON vs. TP800, (**C**) CON vs. TP1200, (**D**) TP400 vs. TP800, (**E**) TP400 vs. TP1200, and (**F**) TP800 vs. TP1200) plots for the four groups of samples. Histograms of rumen differential metabolites in positive and negative ionic states (**G**). Venn diagram showing unique and shared metabolites in positive and negative ionic states (**H**).

**Figure 4 animals-14-02661-f004:**
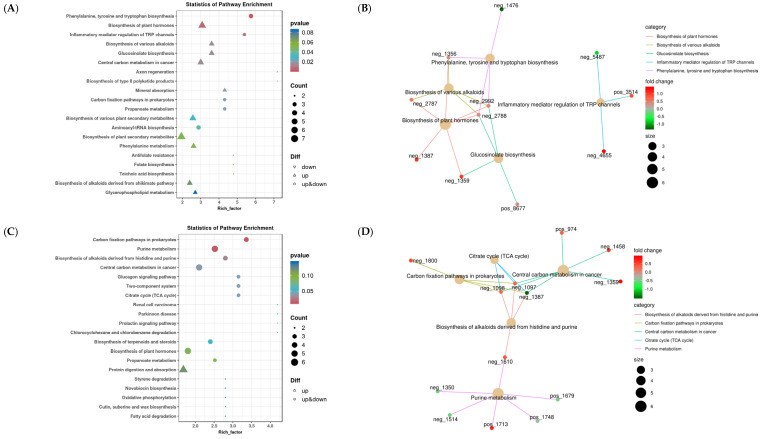
Demonstrates the rumen differential metabolite KEGG enrichment site map and top 5 network map. (**A**–**L**) represent CON vs. TP400, CON vs. TP800, CON vs. TP1200, TP400 vs. TP800, TP400 vs. TP1200, and TP800 vs. TP1200 differential metabolite top 20 enrichment point map and top 5 network map, respectively.

**Figure 5 animals-14-02661-f005:**
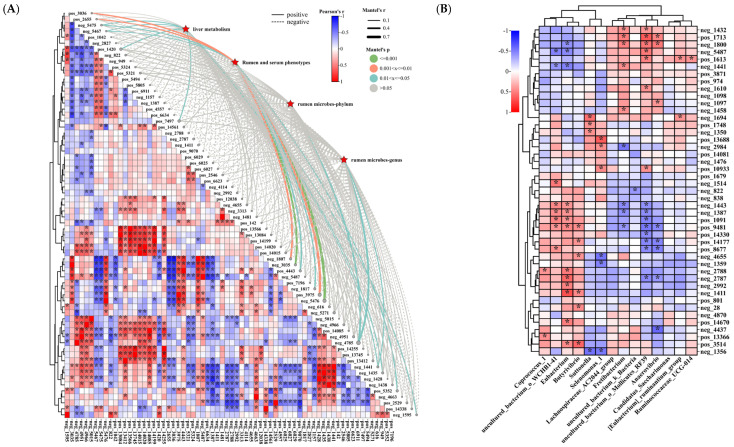
Correlation studies of rumen differential metabolites with liver metabolism, rumen microbiology, rumen parameters, and serum indices in Hu sheep. (**A**) demonstrates Mantel’s r analysis of liver metabolism, rumen and serum phenotypes, rumen microbes/phylum, and rumen microbes/genus with rumen differential metabolites. Spearman’s correlation coefficient magnitude is demonstrated by color change of colored squares. Edge width reflects Mantel’s r statistic for distance correlations, and edge color shows statistical significance from permutations. (**B**) This exhibit shows Spearman’s correlation analysis of rumen biomarkers with rumen parameters and serum indicators. In the heat map, * indicates significant correlation.

**Figure 6 animals-14-02661-f006:**
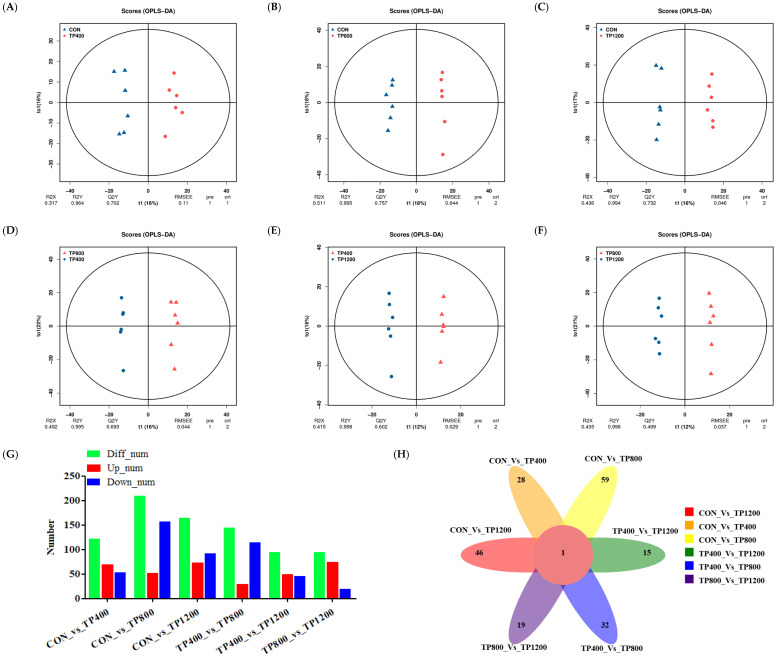
Analysis of liver metabolite differences among the four groups. OPLS−DA ((**A**) CON vs. TP400, (**B**) CON vs. TP800, (**C**) CON vs. TP1200, (**D**) TP400 vs. TP800, (**E**) TP400 vs. TP1200, and (**F**) TP800 vs. TP1200) plots for the four groups of samples. Histograms of rumen differential metabolites in positive and negative ionic states (**G**). Venn diagram showing unique and shared metabolites in positive and negative ionic states (**H**).

**Figure 7 animals-14-02661-f007:**
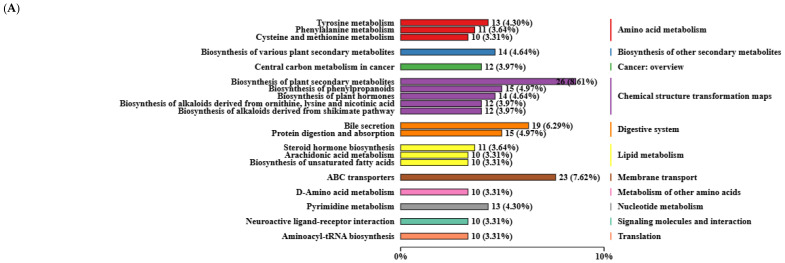
Demonstrates KEGG pathway maps of liver metabolites and top five network maps of differential metabolite enrichment between treatment groups. (**A**) shows KEGG pathway classification maps of liver metabolites. Network diagram of KEGG enrichment of differential metabolites ((**B**) CON vs. TP400, (**C**) CON vs. TP800, (**D**) CON vs. TP1200, (**E**) TP400 vs. TP800, (**F**) TP400 vs. TP1200, and (**G**) TP800 vs. TP1200).

**Figure 8 animals-14-02661-f008:**
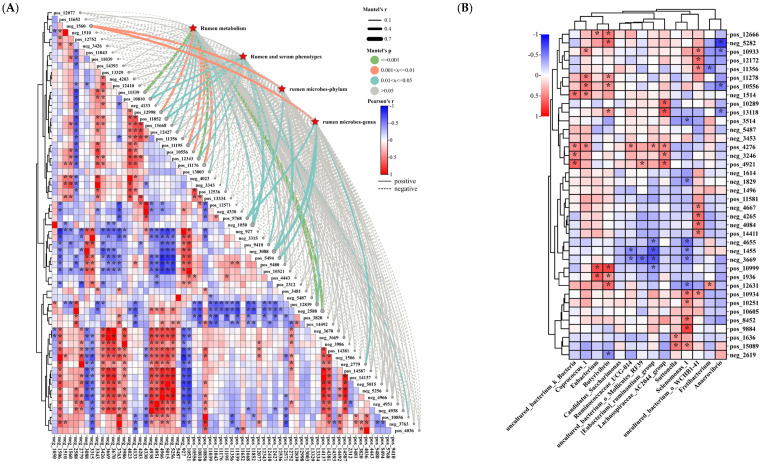
Correlation studies of liver differential metabolites with rumen metabolism, rumen microbiology, rumen parameters, and serum indices in Hu sheep. (**A**) demonstrates Mantel’s r analysis of rumen metabolism, rumen and serum phenotypes, rumen microbes/phylum, and rumen microbes/genus with liver differential metabolites. Spearman’s correlation coefficient magnitude is demonstrated by color change of colored squares. Edge width corresponds to the Mantel’s r statistic for the corresponding distance correlations, and edge color denotes the statistical significance based on permutations. (**B**) exhibits Spearman’s correlation analysis of liver differential biomarkers with rumen parameters and serum indicators. In the heat map, * indicates significant correlation.

**Figure 9 animals-14-02661-f009:**
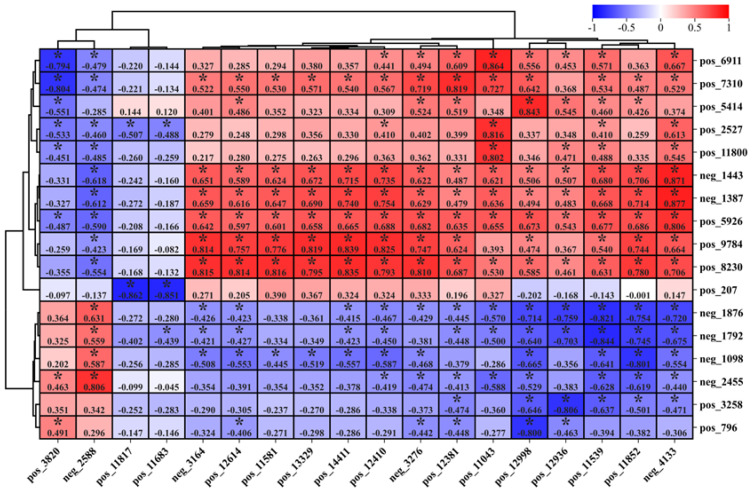
Correlation heat maps of liver and rumen differential metabolites were plotted. In addition, the filtered value of the correlation coefficient was 0.8, while the vertical coordinate was for rumen differential metabolites and the horizontal coordinate was for liver differential metabolites. In the heat map, * indicates a significant correlation.

**Figure 10 animals-14-02661-f010:**
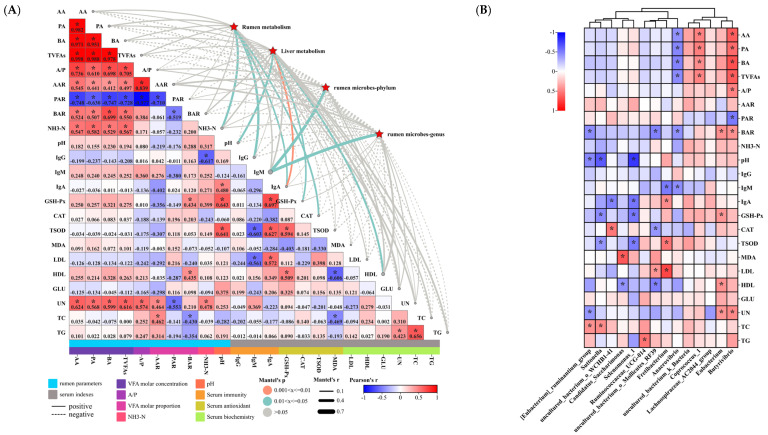
Correlation study of microbiome–metabolome with serum indicators and rumen parameters in Hu sheep. (**A**) demonstrates Mantel’s r analysis of liver metabolism, rumen microbes, and metabolism with rumen parameters (pH, NH_3_−N, AA, PA, BA, TVFAs, AA/PA, AAR, PAR, and BAR) and serum indicators (LDL, HDL, GLU, UN, TC, TG, IgG, IgM, IgA, GSH−Px, CAT, TSOD, and MDA). Spearman’s correlation coefficients are shown numerically in colored squares. Edge width reflects Mantel’s r statistic for distance correlations, and edge color shows statistical significance from permutations. (**B**) Correlation of rumen biomarkers with rumen parameters and serum indicators. In the heat map, * indicates significant correlation.

**Table 1 animals-14-02661-t001:** Ingredients and nutrient levels of basal diet (dry matter basis).

Ingredients	Content (%)	Nutrient Levels ^(2)^	Content (%)
Corn	19.50	DE (MJ/Kg)	11.15
Wheat bran	16.00	CP	16.57
Soybean meal	18.60	NDF	33.14
NaCl	0.40	ADF	24.09
CaCO_3_	0.50	Ca	1.24
Premix ^(1)^	5.00	P	0.44
Peanut vine	40.00		
Total	100.00		

^(1)^ Premix provided the following per kilogram of diet: VA, 100,000 IU; VD3, 15,000 IU; VE, 125 IU; niacin, 250 mg; pantothenic acid, 75 mg; biotin, 5.0 mg; Cu, 50 mg; Zn, 500 mg; Se, 3.75 mg; Fe, 600 mg; I, 8.75 mg; Mn, 500 mg; Co, 3.75 mg. ^(2)^ DE was a calculated value, while other nutrient levels were measured values.

**Table 2 animals-14-02661-t002:** Effect of feed supplementation with TP on serum biochemistry in Hu sheep.

Items	CON	TP400	TP800	TP1200	SEM	*p* Value		
						ANOVA	Linear	Quadratic
LDL, μmol/L	242.05	241.88	267.06	271.96	7.50	0.384	0.099	0.866
HDL, μmol/L	49.13	55.04	50.16	53.35	0.89	0.056	0.287	0.403
GLU, mmol/L	2.83	3.48	3.30	3.18	0.13	0.377	0.473	0.159
UN, mmol/L	0.12	0.13	0.12	0.11	0.01	0.665	0.538	0.492
TC, mmol/L	0.20	0.17	0.15	0.17	0.01	0.382	0.251	0.226
TG, mmol/L	2.70	2.62	2.32	3.06	0.22	0.726	0.707	0.384

CON: control group; TP400: the basal diet was supplemented with 400 mg/kg of tea polyphenols; TP800: the basal diet was supplemented with 800 mg/kg of tea polyphenols; TP1200: the basal diet was supplemented with 1200 mg/kg of tea polyphenols; SEM: standard error of mean; LDL: low−density lipoprotein; HDL: high−density lipoprotein; GLU: glucose; UN: urea nitrogen; TC: total cholesterol; TG: triglyceride.

**Table 3 animals-14-02661-t003:** Effect of feed supplementation with TPs on serum immunity and antioxidants in Hu sheep.

Items	CON	TP400	TP800	TP1200	SEM	*p* Value		
						ANOVA	Linear	Quadratic
IgG, ug/mL	2903.28	3056.56	3253.38	2903.28	105.75	0.758	0.842	0.261
IgM, μg/mL	39.71	36.41	36.41	37.11	1.71	0.457	0.646	0.598
IgA, μg/mL	23.65 ^c^	31.46 ^b^	34.97 ^ab^	41.61 ^a^	1.56	<0.001	<0.001	0.732
GSH−Px, pmol/mL	88.70 ^c^	102.79 ^ab^	95.64 ^bc^	105.74 ^a^	1.48	<0.001	<0.001	0.102
CAT, ng/L	474.52 ^ab^	549.76 ^a^	378.98 ^c^	432.48 ^bc^	15.68	0.002	0.002	0.569
TSOD, pg/mL	204.12 ^b^	240.05 ^a^	233.97 ^a^	254.02 ^a^	5.12	0.003	<0.001	0.294
MDA, nmol/mL	18.60	15.72	17.94	16.00	0.58	0.053	0.271	0.678

CON: control group; TP400: the basal diet was supplemented with 400 mg/kg of tea polyphenols; TP800: the basal diet was supplemented with 800 mg/kg of tea polyphenols; TP1200: the basal diet was supplemented with 1200 mg/kg of tea polyphenols; SEM: standard error of mean; IgG: immunoglobulin G; IgM: immunoglobulin M; IgA: immunoglobulin A; GSH−Px: glutathione peroxidase; CAT: catalase; TSOD: total superoxide dismutase; and MDA: malondialdehyde. Peer numbers without shoulder letters or with the same letter indicate non−significant differences, whereas different letters indicate significant differences.

**Table 4 animals-14-02661-t004:** Effect of feed supplementation with TPs on rumen fermentation in Hu sheep.

Items	CON	TP400	TP800	TP1200	SEM	*p* Value		
						ANOVA	Linear	Quadratic
pH	7.38 ^b^	7.57 ^a^	7.56 ^a^	7.55 ^a^	0.02	<0.001	<0.001	<0.001
NH_3_−N	23.22	25.65	23.89	26.55	0.76	0.398	0.238	0.940
VFA molar concentration (mmol/L)								
AA	14.90	17.37	11.96	17.26	1.19	0.348	0.875	0.555
PA	4.01	4.49	3.11	4.54	0.50	0.216	0.926	0.377
BA	2.75	3.74	2.26	3.40	0.26	0.195	0.841	0.883
TVFAs	23.59	27.94	19.05	27.66	3.38	0.284	0.838	0.559
AA/PA	3.72	3.71	3.72	3.73	0.12	1.000	0.936	0.935
VFA molar proportion								
AAR	63.23	61.41	62.09	61.73	0.85	0.320	0.242	0.318
PAR	17.03	16.74	16.82	16.63	0.34	0.924	0.561	0.921
BAR	11.34 ^b^	13.04 ^a^	11.65 ^b^	12.22 ^ab^	0.23	0.047	0.513	0.192

CON: control group; TP400: the basal diet was supplemented with 400 mg/kg of tea polyphenols; TP800: the basal diet was supplemented with 800 mg/kg of tea polyphenols; TP1200: the basal diet was supplemented with 1200 mg/kg of tea polyphenols; SEM: standard error of mean; NH_3_−N: ammonia nitrogen; VFA: volatile fatty acid; acetic acid: AA; propionic acid: PA; butyric acid: BA; acetic acid/propionic acid: AA/PA. Peer numbers without shoulder letters or with the same letter indicate non−significant differences, whereas different letters indicate significant differences.

**Table 5 animals-14-02661-t005:** Effect of feed supplementation with TPs on rumen microbial alpha diversity in Hu sheep.

Items	CON	TP400	TP800	TP1200	SEM	*p* Value
ACE	817.14	789.96	824.73	821.02	16.28	0.884
Chao1	813.52	790.63	824.13	813.80	15.88	0.906
Simpson	0.98	0.98	0.98	0.98	0.00	0.411
Shannon	7.39	7.53	7.35	7.63	0.08	0.644

CON: control group; TP400: the basal diet was supplemented with 400 mg/kg of tea polyphenols; TP800: the basal diet was supplemented with 800 mg/kg of tea polyphenols; TP1200: the basal diet was supplemented with 1200 mg/kg of tea polyphenols; SEM: standard error of mean.

## Data Availability

Available by contacting the author.

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
