# Peer review of "Microbial–Metabolomic Exploration of Tea Polyphenols in the Regulation of Serum Indicators, Liver Metabolism, Rumen Microorganisms, and Metabolism in Hu Sheep"

_animals, 2024, doi:10.3390/ani14182661_

Round 1

Reviewer 1 Report

Comments and Suggestions for Authors

1.The scientific problem is unclear, and the author studied the effects of different levels of TP addition on a series of indicators, but there is very little description of dose-response in the article. And no clear viewpoint was given in the conclusion

2.The writing logic is chaotic and the structure is poor.

3. The language expression is not concise.

4. There are still many detailed issues in this article that need to be checked and revised.

Comments on the Quality of English Language

Minor editing of English language required.

Author Response

Thank you very much for your valuable suggestions to improve our manuscript. These comments have been valuable in revising and improving our manuscript and have been an important guide for our research. We have carefully reviewed the comments and made corrections that we hope will be accepted. The changes to our manuscript within the document were also highlighted by using blue colored text. Enclosed, please find point-by-point responses addressing all comments below.

Reviewer 2 Report

Comments and Suggestions for Authors

Comments and Suggestions for Authors

After reviewing the manuscript entitled “Microbial-Metabolomic Exploration of Tea Polyphenols on the Regulation of Serum Indicators, Liver Metabolism, Rumen Microorganisms and Metabolism in Hu sheep”, the following suggestions were made it. The manuscript is interesting and provides novel information on the use of polyphenols in ruminal microbiota and metabolism in lambs. However, several weaknesses need to be improved before the manuscript can be considered for publication. Probably the weakest sections are the introduction and conclusions, as they are not clear and do not provide enough information. Below are my specific comments:

Simple Summary and Abstract

Lines 21-22: The Simple Summary and Abstract are well structured and written. However, abbreviations should not be used in this section. Please correct.

Lines 25-26: The conclusion needs to be clarified and corrected. As it stands, it mentions that the manuscript's findings provide "valuable information" on the regulatory effects of tea polyphenols. However, this conclusion is too ambiguous, and the authors need to rewrite it specifically. Considering their regulatory mechanisms, it should be clearly stated what kind of effects tea polyphenols have.

Keywords: tea polyphenols; serum indicators; rumen microorganisms; liver metabolism. These words used as keywords are the same as those previously used in the title of the manuscript. Keywords should be different from those in the title (but related to the topic) to broaden the reach of academic search engines in case the manuscript is later published.

Introduction

Lines 48-51: Are there scientific references to support these claims?

Lines 52-54: Plant bioactive compounds can indeed replace antibiotics in many functions; however, it is unclear what problem they are trying to solve. The authors should add more information that clearly shows the main uses of antibiotics and the main problems they cause. This is important so the reader understands the importance of using plant-derived products instead of antibiotics.

Lines 58-59: Authors should adequately justify why they recommend using tea polyphenols instead of other plant-derived bioactive compounds.

Line 63: The word “flora” is incorrect and should be replaced by microbiota. This correction should be applied throughout the manuscript.

Lines 92-94: Are there scientific references to support these claims?

Material and methods

Line 113: Authors should add the average weight (and its standard error) of the lambs used. They should also indicate the average age of the animals.

Line 116: A scientific justification should be added explaining why the doses of 400, 800, and 1200 mg/kg DM of tea polyphenols were used.

Line 190: Were statistical tests used to assess the normality of the data prior to the statistical analysis? If they were not used, the analyses should be performed and changed (in case the tests show that the data are not normal). Also, the authors should add the statistical model used to assess the different groups of response variables.

Results

Line 209: Please delete this line as it shows repetitive and irrelevant information.

Table 2: Abbreviations CON, TP400, TP800, TP1200, SEM, LDL, HDL, GLU, UN, TC, TG, IgG, IgM, IgA, GSH-Px, CAT, TSOD, and MDA must be defined in the footer of Table 2.

Table 3: Abbreviations CON, TP400, TP800, TP1200, SEM, NH3-N, and VFA must be defined in the footer of Table 3.

Line 278: Change “Table 3” to “Table 4”.

Lines 278-284: The authors should be more specific in the description of the results and should add the significance value (p value) with which statistical differences were detected or not.

Table 4: Abbreviations CON, TP400, TP800, and TP1200 must be defined in the footer of Table 3.

Lines 299-340, and 354-366: The authors should be more specific in the description of the results and should add the significance value (p value) with which statistical differences were detected or not.

Lines 410-432, 443-454, and 544-554: The authors should be more specific in the description of the results and should add the significance value (p value) with which statistical differences were detected or not.

Discussion

Line 561: Change “microflora” to “microbiota”.

Lines 579-582: The discussion of antioxidant enzymes is too ambiguous and needs to be improved. It is not enough to contrast the results obtained with the findings of previous studies. The authors should use tea polyphenols' biochemical and physiological mechanisms to explain why they increased serum levels of IgA, GSH-Px, and TSOD in the current study.

Lines 585-586: It is unclear how the positive effects of tea polyphenols on rumen modulation and hepatic metabolism are related to improved immune response and antioxidant capacity. Therefore, the authors should clarify this hypothesis using scientific references as support.

Lines 587-588: The discussion section should not describe what was measured in the study. Therefore, these two lines should be removed.

Lines 695-699: These lines should be removed as they only briefly repeat the previously obtained results described in the results section.

Conclusions

Lines 701-706: The conclusions section should be completely rewritten because it only briefly restates the findings obtained in their current form. A good conclusion should briefly and concisely state the implications and applications of the results obtained without repeating them.

Lines 707-709: It is not enough to mention that it regulates hepatic metabolism because it influences metabolic pathways. It is necessary to specify which metabolic pathways and in what sense (increase or decrease) they are modified. Likewise, based on the reviewed background, it must be concluded by considering the possible implications of the changes detected in the metabolic pathways.

Comments on the Quality of English Language

The quality of the English language is poor, with several sections being difficult to understand. The main problems are the use of overly long sentences, which make understanding difficult, and the use of incorrect grammatical structures.

Author Response

(The authors gave the same response as above.)

Round 2

Reviewer 1 Report

Comments and Suggestions for Authors

L126. I have doubts about the weight of the experimental animals, and there is a lack of units here.

L233. The scope of the term 'serum' is too broad, it is suggested that the title be more specific.

L269. The author analyzed rumen fluid samples using 16S sequencing, but the results do not seem to represent all microorganisms in the rumen, only including data on bacteria.

L752. Please review the conclusion for clarity, these sections require English revision to be suitable for publication. In addition, in this section, the author suggests that the addition amount between 400-800mg/kg may have beneficial effects on serum indicators, but the optimal addition amount has not yet been proposed. This may not have guiding significance for actual production.

Comments on the Quality of English Language

Minor editing of English language required. The abstract and conclusion sections need to be emphasized to meet the publishing requirements.

Author Response

(The authors gave the same response as above.)

Reviewer 2 Report

Comments and Suggestions for Authors

Comments and Suggestions for Authors

After reviewing the manuscript entitled “Microbial-Metabolomic Exploration of Tea Polyphenols on the Regulation of Serum Indicators, Liver Metabolism, Rumen Microorganisms and Metabolism in Hu sheep”, the following suggestions were made it. The authors made a great effort to respond appropriately to each of my comments and corrections. I only have a couple of additional minor corrections.

Comment 1. All abbreviations included in Table 2 must be defined in the table caption.

Comment 2. In the response variables in Table 3, the linear and quadratic effects must be estimated and added.

Comments on the Quality of English Language

The quality of the English language is good, no major changes are required.

Author Response

(The authors gave the same response as above.)
